# Effect of the Drying Method and Optimization of Extraction on Antioxidant Activity and Phenolic of Rose Petals

**DOI:** 10.3390/antiox12030681

**Published:** 2023-03-10

**Authors:** Sabrina Baibuch, Paula Zema, Evelyn Bonifazi, Gabriela Cabrera, Alicia del Carmen Mondragón Portocarrero, Carmen Campos, Laura Malec

**Affiliations:** 1Industries Department, Faculty of Exact and Natural Sciences, University City, University of Buenos Aires, Buenos Aires C1428EGA, Argentina; 2Organic Chemistry Department, Faculty of Exact and Natural Sciences, University City, University of Buenos Aires, Buenos Aires C1428EGA, Argentina; 3Institute of Food Technology and Chemical Processes, Faculty of Exact and Natural Sciences, University City, National Scientific and Technical Research Council, Buenos Aires C1428EGA, Argentina; 4Microanalysis and Physical Methods Applied to Organic Chemistry Unit, University City, National Scientific and Technical Research Council, Buenos Aires C1428EGA, Argentina; 5Department of Analytical Chemistry, Nutrition and Food Science, Faculty of Sciences Veterinary, University of Santiago de Compostela, 27002 Lugo, Spain

**Keywords:** rose petals, phenolic compounds, antioxidant activity, response surface methodology, modeling and optimization, ultrasound-assisted extraction

## Abstract

The effect of freeze and hot air drying methods on the retention of total phenolics, antioxidant activity (AA), and color of different cultivars of rose petals was analyzed. Both methods similarly preserved the phenolic content and AA, while freeze drying showed better red color retention. Furthermore, the conditions of total phenolics and AA extraction from two rose cultivars, *Lovely Red* and *Malu*, were optimized by response surface methodology through a Box–Behnken design. The solvent exhibited a major effect on the total phenolic content (TPC) and AA. The selected parameters were ethanol 38%, 75 °C, and 30 min. Under these conditions, the predicted values for *Lovely Red* were 189.3 mg GA/g dw (TPC) and 535.6 mg Trolox/g dw (AA), and those for *Malu* were 108.5 mg GA/g dw (TPC) and 320.7 mg Trolox/g dw (AA). The experimental values were close to the predicted values, demonstrating the suitability of the model. Ultrasound-assisted extraction increased the AA of the extracts but not the TPC. Fifteen compounds were identified in the *Lovely Red* cultivar, with no differences between the two drying methods. The results obtained suggest that the analyzed cultivars, particularly the red ones, can be considered a natural source of powerful antioxidant compounds.

## 1. Introduction

Numerous studies have demonstrated that flower petals contain high levels of phenolic compounds possessing antioxidant activity (AA) [1,2,3,4]. In particular, petals of some rose cultivars are known to contain the highest antioxidant capacity amongst other flowers [1]. However, information about the total phenolic content (TPC) and AA of roses commonly cultivated in Argentina is scarce. In northeastern Buenos Aires, rose cultivation occupies a strategic position in productive and commercial organizations [5]. When the plant reaches a salable size, it produces several blooms. These early flowers are discarded, becoming waste. However, on account of their TPC and antioxidant properties, these waste products could have potential industrial uses as nutraceuticals and preservatives. Currently, synthetic antioxidants are often applied in different lipid-rich matrices to prevent lipid oxidation. However, their use is controversial because they can cause or promote negative health effects [6]. For this reason, there is an increase in the demand for antioxidants from natural sources as substitutes for synthetic ones [7] Among natural compounds, phenolics are a promising option to prevent lipid oxidation of unsaturated fatty acids in food oils. In particular, it was reported that extracts of edible flowers enhanced oxidative stability of the cold-pressed chia and flax seed oils [8].

The stability and availability of phenolic compounds and AA in rose petals are related to the drying and extraction conditions. Drying is an important process for handling food products, as well as for inhibiting enzymatic reactions, preventing microbial growth, and reducing weight for cheaper transport and storage. Drying conditions, mainly the temperature and time applied, affect the composition of the sample obtained [9]. The methods of freeze drying and hot air drying are commonly used for food dehydration. To prevent the degradation of bioactive compounds, the use of temperatures below 70 °C is necessary when drying by hot air. Freeze drying has been shown to be effective in the preservation of bioactive compounds from flowers. For example, black locust flowers processed by freeze drying exhibited higher AA than those processed by sun drying, hot air drying, or vacuum-microwave drying [10]. However, freeze drying is time consuming and expensive.

Solid–liquid extraction is one of the most widely used procedures for the extraction of phenolic compounds and AA [7]. The solvent, temperature, extraction time and solid-to-liquid ratio are the variables that control the extraction process [11]. Optimization of extraction conditions is essential to maximize the isolation of phenolic compounds and AA [12]. To determine these conditions, it is convenient to use an experimental design to reduce the number of experiments carried out and to predict untested combinations. Response surface methodology allows the evaluation of the effects of multiple factors and their interactions. In this way, the optimal extraction conditions for the tested process can be predicted. Although a few previous studies analyzed the extraction conditions of phenolics from rose petals [13,14], most of them evaluated one factor at a time.

At present, there is a particular interest in using extraction methods based on environmentally friendly processes. Along these lines, the use of aqueous ethanol and ultrasound-assisted extraction have been successfully applied to the extraction of phenolic compounds from different vegetable matrices [15].

Based on the promising phenolic content and antioxidant activity of rose petal extracts that could be used as natural additives to prevent lipid oxidation in food products, this study aims to evaluate the effect of freeze drying and hot air drying on total phenolics, AA and the color of the obtained powders from different Argentinian rose cultivars. Furthermore, the extraction conditions of total phenolics and AA in two selected cultivars were optimized by employing a Box–Behnken design. Following selection of the optimal conditions, the impact of ultrasound-assisted extraction was also evaluated. Finally, the phenolic composition of hot air- and freeze-dried petals of a selected cultivar analyzed by HPLC-ESI-QTOF/MS were compared.

## 2. Materials and Methods

### 2.1. Chemicals

Folin–Ciocalteu reagent and gallic acid (GA) were purchased from Merck (Darmstadt, Germany), and6-hydroxy-2,5,7,8-tetramethylchromane-2-carboxylic acid (Trolox) and 2,2-diphenyl-1-picryl-hydrazyl (DPPH) were purchased from Sigma (Steinhelm, Germany). Cyanidin-3-O-glucoside standards was supplied by Fluka (St. Louis, MO, United States). Absolute ethanol was purchased from Biopack (Buenos Aires, Argentina). All other chemicals used were of analytical grade.

### 2.2. Plant Material

Fresh rose (*Rosa* spp.) flowers free of pesticides were collected in the nurseries of San Pedro, Buenos Aires, Argentina, in December 2018. Petals of 40 flowers from four different cultivars were used: *Gran Gala* (GG), *Traviata* (T), *Lovely Red* (LR) (red color), and *Malu* (M) (pink color). The rose petals were sampled randomly and manually separated.

### 2.3. Drying Treatments

The rose petals of the four cultivars were spread in thin layers in separate trays for drying treatments. Freeze drying was carried out at 4 Pa and −55 °C for 48 h (Martin Christ Alpha 1–4 LD, Osterode am Harz, Germany). Hot air drying conditions were selected by preliminary tests and were performed at 65 °C for 1.3 h using a hot circulation oven (Edgware-Middlesex M008H104, Edgware-Middlesex, United Kingdom). Afterwards, the dried petals were ground to a fine powder. Particles of 0.850 mm were separated using laboratory sieves (No. 20 A.S.T.M.) and stored at −18 °C in amber colored glass containers until analysis. The moisture content of the plant material was determined in a vacuum oven at 65 °C up to constant weight.

### 2.4. Color Measurement

The color of the dehydrated powders was measured with a tristimulus colorimeter (Konica Minolta CM-600D, Tokyo, Japan) with a D65 illuminant and a 2° observer angle, previously calibrated in the CIE Lab color space. The display was set to the CIELAB scale, with L*, a*, and b* coordinates, where L* varies between 0 (black) and 100 (white), a* varies from green (−) to red (+), and b* varies from blue (−) to yellow (+). The reported values were the average of three readings.

### 2.5. Anthocyanin Content

Total content of anthocyanins (TA) was determined in triplicate using the pH differential method according to Giusti and Wrolstad (2001) [16]. Two dilutions were prepared of the sample, one with potassium chloride buffer, pH 1.0, and the others with sodium acetate buffer, pH 4.5. After 15 min, the absorbance of each solution was measured at 510 and 700 nm. Cyd-3-glu was used as a reference standard.

### 2.6. Extraction Procedure

The dried powder of rose petals (0.02–0.04 g) was extracted with a mixture of different proportions of distilled water and absolute ethanol. These solvents have been rated by the FDA as generally recognized as safe (GRAS) substances. In addition, they are environmentally friendly. The solid-to-solvent ratio used was in the range of 1:250 to 1:500 g/mL. These ratios are lower than those used in other studies for the extraction of phenolics from flower petals [1,2]. Therefore, this ratio was considered sufficient for the efficient recovery of phenolics and AA. To evaluate the drying process, the extraction of freeze-dried and hot air-dried samples was carried out in a thermostatic bath using 50% ethanol at 50 °C for 15 min with ultrasound assistance. Then, optimization of the extraction was performed using freeze-dried samples of the M and LR cultivars. Different solvent, temperature, and time conditions were applied as detailed in the experimental design (Table 1). The extract was centrifuged (Eppendorf AG DCS-16RV R, Hamburg, Germany) at 6440× *g* for 10 min at 4 °C. The pellet was resuspended in the solvent and then centrifuged under the same conditions. The two supernatants were combined into 10 mL with solvent and stored at −18 °C until evaluation of the AA and TPC. The extractions were performed in duplicate.

### 2.7. Measurement of Total Phenolic Content

The TPC was determined using the Folin–Ciocalteu method with slight modifications [17]. An aliquot (0.25 mL) of appropriately diluted extract was added to 4.0 mL of distilled water and 0.25 mL of Folin–Ciocalteu reagent. After 3 min, saturated sodium carbonate solution (0.5 mL) was added. The absorbance was measured at 750 nm using a UV/Vis Lambda 25 spectrophotometer (Perkin Elmer, Waltham, MA, United States) after incubation at 30 °C for 2 h. Gallic acid was used as a reference standard, and the results were expressed as mg GA per g dry weight (dw) petals. Determinations were performed in triplicate.

### 2.8. Measurement of Antioxidant Activity

Antioxidant activity was measured using the DPPH method of Brand-Williams et al. with slight modifications [18]. An aliquot of 400 μL of diluted extract was added to 3.6 mL of 0.1 mM DPPH solution. After incubation in the dark at 30 °C for 30 min, the absorbance was read at 517 nm. The AA of the extracts was evaluated in triplicate, and the results were expressed as mg Trolox per g dw.

### 2.9. Experimental Design

To optimize the extraction conditions of TPC and AA from rose cultivars, a Box–Behnken design was used. The factors evaluated were ethanol percentage, temperature, and extraction time. The design was organized into two blocks, and each factor was evaluated at two levels. Three center points per block were used for a total of 30 experiments per cultivar. The order of the experiments was randomized. Table 1 details the levels used. The experimental data for the two responses (TPC and AA) were fitted to a second-order polynomial model, with linear, quadratic, and interaction of the variable terms:$$Y=\beta_{0}+\overset{3}{\underset{i=1}{\sum }}\beta_{i}X_{i}+\overset{3}{\underset{i=1}{\sum }}\beta_{ii}X_{i}{}^{2}+\overset{3}{\underset{i=1}{\sum }}\overset{3}{\underset{j=1}{\sum }}\beta_{ij}X_{i}X_{j}$$
where Y is the response, $\beta_{0}$ is the coefficient of the intercept term, $\beta_{i}$ are the coefficients of the linear terms, $\beta_{ii}$ are the coefficients of the quadratic terms, $\beta_{ij}$ are the coefficients of the interaction terms, and $X_{i}$ and $X_{j}$ are the independent variables.

The predictive ability of the developed model was validated by additional extractions from the LR cultivar using 50% ethanol at 90 °C for 30 min.

### 2.10. Effect of Ultrasound-Assisted Extraction

The effect of ultrasound-assisted extraction was analyzed under the following conditions: 38% ethanol at 75 °C for 30 min with and without ultrasound assistance. Ultrasound-assisted extraction was performed in an ultrasonic bath (frequency of 37 kHz; Elmasonic S 30 H, Elma, Singen, Germany).

### 2.11. HPLC-ESI-QTOF/MS Analysis

The phenolic composition of LR cultivar extracts from hot air- and freeze-dried samples submitted to the most convenient selected conditions was analyzed by HPLC-ESI-QTOF/MS with an Agilent 1200 HPLC (Agilent Technologies, Wilmington, DE, United States). The HPLC equipment was coupled to a high-resolution mass spectrometer Bruker QTOF-QII (Bruker Daltonics, 276 Billerica, MA, United States) with an electrospray ionization source (ESI). The ionization conditions were a capillary temperature of 200 °C and a voltage of 4.5 kV. The pressure of nitrogen as the nebulizer gas was 3.0 bar, and its flow rate as the drying gas was 6.0 L/min. The mass scan was performed between 100 and 1000 *m*/*z* in positive- and negative-ion modes. The column used was a Phenomenex Luna^®^ 3 μm C18(2) 100 Å, 100 × 2 mm, with a mobile phase of 0.1% *v*/*v* formic acid (A) and methanol (B), using a 0.3 mL/min flow rate and 5 μL injection volume.

### 2.12. Data Analysis

Experimental data from the drying process were analyzed using one-way analysis of variance (ANOVA) and Tukey’s test. For extraction optimization, the significance of variables and their interactions was evaluated using ANOVA. The determination coefficient (R^2^), the adjusted coefficient of determination (R^2^ adj), and the lack-of fit test were calculated to evaluate the model adequacy. The conditions for the extraction of maximal TPC and AA were determined by the desirability function method. The correlation between TPC and AA values was analyzed using simple linear regression. HPLC-ESI-QTOF/MS data were acquired and processed using the Bruker Compass Data Analysis ver. 4.2 (Bruker Daltonics, Billerica, MA, United States).

The statistical significance level value was set at the 5% level (*p* value < 0.05), unless otherwise mentioned. All analyses were performed using the Statgraphics program (Statgraphics Centurión XV, version 15.02.05, Statpoint, Inc., The Plains, VA, USA).

## 3. Results and Discussion

### 3.1. Effect of the Drying Process on TPC, AA, Anthocyanins and Color

Figure 1 shows the effect of freeze drying and hot air drying on the TPC (a), AA (b), and a* parameter (c) of the GG, T, LR, and M cultivars. For TPC, there were no significant differences between the two drying methods in GG and LR; however, in M, the values obtained by hot air drying were higher, and in T, those obtained by freeze drying were higher. In the hot air-dried samples, it the AA was higher than that of the freeze-dried samples in GG, and M, while in LR and T, it was lower. Previous studies have shown different trends about TPC and AA depending on the type of flower analyzed. Freeze drying was the best method for preserving the TPC and AA on Jasminum species and Common Daisy compared to hot air drying [19,20]. However, for Lavender flowers hot air-dried samples had higher phenolic content than freeze-dried and similar AA for both drying methods [20].

Regarding anthocyanin content, it was in decreasing order: 5.09 ± 0.09, 3.88 ± 0.42, 2.41 ± 0.23, 0.98 ± 0.05 mg cyd-3-glu/g ms, for LR, GG, T and M, respectively. As expected, the lower content was found for the pink cultivar M. To evaluate, the possible contribution of anthocyanin content to the AA, its ratio to phenolic content was estimated in molar basis and it was found that the proportion was lower than 1% for the four cultivars. Therefore, according to these estimations, the contribution to anthocyanins to the total antioxidant activity in these samples could be considered not relevant. For this reason, we have focused on optimizing the AA and TPC in the extracts.

Color was analyzed according to the parameter a*, as it characterizes the proportion of red in the samples. The highest red color retention was achieved by freeze drying, since it presented the highest a* values in the four cultivars. One of the main benefits of freeze-drying flowers is the retention of fresh color, since higher temperatures promoted color changes [21]. The abovementioned trends were not conclusive in selecting between these two drying methods, and both could be considered suitable for preserving the TPC and AA of rose petals. Degradation of pigments might be related to the loss of some bioactive compounds. Hence, regarding the retention of red color, freeze drying would be advisable. On account of that, freeze-drying was applied in the analyses carried to select the extraction conditions.

### 3.2. Optimization of the Extraction Process

#### 3.2.1. Analysis of the Model

Response surface methodology was used to optimize the extraction of total phenolics and AA from rose petals. Table 2 details the values of TPC and AA experimentally obtained for cultivars LR and M under the different conditions according to the experimental design. The pink cultivar M was selected for optimization with relatively low TPC and AA. On the other hand, LR showed intermediate results among the red cultivars. TPC were within the range of 28.1 to 173.9 mg GA/g dw for LR and of 15.2 to 118.4 mg GA/g dw for M. The AA varied from 65.0 to 581.8 mg Trolox/g dw for LR and from 25.5 to 359.3 mg Trolox/g dw for M. The wide range of TPC and AA values obtained suggests that at least one of the variables studied exerted a profound influence on the extraction efficiency. It has also been observed that the red petals (LR) presented higher AA and TPC in all the conditions tested, compared with the pink cultivar (M).

Table 3 shows the ANOVA results. The *p* values of the lack of fit indicate that the variables analyzed were sufficient to explain the proposed model. In the case of TPC for LR, the low value of lack of fit could be attributed to the nonsignificant effect of time on total phenolic extraction for this cultivar. Therefore, the total phenolic response for LR depended only on the ethanol % and temperature. The R^2^ adj for TPC and AA indicated that the model explained 96–98% of the variability of both responses. The R^2^ was close to the R^2^ adj, confirming the accuracy of the model. Thus, the model was appropriate to predict the best extraction conditions within the tested range.

#### 3.2.2. Effect of the Different Variables on TPC and AA Extraction

The level of significance for the effect of each variable and their interactions can be observed in Table 3. The solvent exhibited a highly significant influence on the extraction of total phenolics and AA, which is expressed in the linear (X_1_) and quadratic (X_1_^2^) terms for LR and M. The temperature also affected TPC and AA in its linear (X_2_) expression. However, the quadratic term (X_2_^2^) was only significant for AA for both cultivars. Time showed a significant effect in the linear term (X_3_) on TPC and AA for the M cultivar and in the quadratic term (X_3_^2^) on AA for the LR cultivar. The interaction between variables was not significant, except for solvent and temperature (X_1_ X_2_) on TPC for M. The regression coefficients were obtained from the experimental design calculated by the desirability function method. Following removal of nonsignificant terms, the model equations for TPC and AA in the extracts could be described by the following quadratic polynomial equations:TPC *LR* = 163.92 − 38.32 X_1_ + 12.59 X_2_ − 76.22 X_1_^2^
TPC *M* = 104.18 − 27.24 X_1_ + 9.26 X_2_ − 3.67 X_3_ − 47.09 X_1_^2^ + 4.49 X_1_ X_2_
AA *LR* = 445.96 − 127.70 X_1_ + 45.91 X_2_ − 233.88 X_1_^2^ + 61.80 X_2_^2^ + 37.60 X_3_^2^
AA *M* = 303.69 − 76.46 X_1_ + 36.14 X_2_ − 13.07 X_3_ − 185.72 X_1_^2^ + 18.00 X_2_^2^

Predictive equations were used to generate three-dimensional response surface curves, which were plotted by varying two variables and keeping the third constant at the central point. The effects of ethanol % and temperature at 75 min on TPC and AA are shown for LR in Figure 2a,c and for M in Figure 2b,d. The effects of ethanol % and time at 60 °C on TPC and AA are presented in Figure 3a,c for LR and Figure 3b,d for M. All these figures show that the solvent percentage presented a maximum value for the extraction of phenolics and AA, as also expressed by the negative value of the quadratic term of the equations. Moreover, the negative linear term of the solvent means that for the maximum extraction conditions, the solvent contains a greater proportion of water. Pure ethanol showed the lowest extraction ability for M and LR. Phenolics are predominantly hydrophilic compounds able to interact through hydrogen bonds with the solvent and therefore exhibiting better solvation with water. On the other hand, extraction of rose petals with pure water did not completely extract all polyphenols, as stated by Chew et al. [22]. Thus, binary systems (water/ethanol) allowed for greater extraction of phenolic compounds. These results are in agreement with those of Trinh et al. [14], who reported higher phenolic contents in hydroalcoholic extracts from roses. Phenolics are usually found in the matrix of plants and are linked by hydrogen bonds or hydrophobic bonds to proteins and/or polysaccharides. It has been reported that in rose flowers, there is a significant proportion of carbohydrates as well as proteins and lipids [23], where polyphenols could possibly be attached. Most likely, binary systems are better able to break these bonds and thus release polyphenols.

Increasing the temperature improved the extraction, according to the positive linear relation between temperature and TPC and AA for both cultivars. High temperature increases the diffusion coefficient, facilitates solvent penetration, and enhances solubility. In addition, it can help to breakdown the bonds between phenols and proteins or polysaccharides, facilitating the release of phenolics. However, the coefficients of the temperature terms in all cases were considerably lower than those of the solvent terms, explaining the smaller effect of this parameter, as shown in Figure 2.

The negligible influence of the extraction time observed on TPC and AA (Figure 3) agreed with the lowest coefficient of this variable in the equations. Furthermore, this trend was expected since rapid extraction of solutes takes place in the initial minutes until equilibrium is reached between the solute and the solvent (Fick’s second law of diffusion). Subsequently, increasing the time does not favor further extraction but could promote oxidation reactions.

The optimal conditions of solvent and temperature for phenolic compounds and AA extraction were the same for LR and M, while the time was in the range of 112–120 min for both cultivars (Table 4). Consequently, 116 min was selected as an average time. It must be stressed that the differences between the predicted TPC and AA for this time and for each optimal time were lower than 1.5%.

As mentioned above, elevated temperature of extraction favored a higher recovery of TPC and AA for both cultivars. However, extraction at high temperatures for a long time could cause deterioration of phenolic compounds by thermal degradation [24]. Thus, this latter effect could counteract the higher extraction promoted by the increase in temperature. Furthermore, the use of elevated temperatures implies high energy consumption. For the mentioned reasons and considering the slight effect observed for the extraction time, it was of interest to compare the responses using milder time–temperature extraction conditions, such as 75 °C—30 min, and keeping the optimal solvent percentage. It was found that predicted values at 75 °C—30 min for the TPC of both cultivars were close to the values for the optimal conditions (ethanol 38%, 90 °C, 116 min), while the AA was lower (Table 4). Most likely, some decomposition products formed at higher temperature–time conditions would still exhibit radical-scavenging activity, as was previously reported for some phenolic compounds by Murakami et al. [25]. Even though a greater AA could be extracted at 90 °C, 75 °C was selected as the most convenient temperature, taking into account the possible damage of some labile phenolic compounds as well as the lower energetic cost of the process. Thus, 38% ethanol, 75 °C and 30 min were the selected conditions of % ethanol, temperature and time for the extraction of phenolic compounds and antioxidant activity from rose petals. The TPC and AA obtained from LR under such conditions are comparable to the highest values previously reported for other flowers or even other rose cultivars [14,23,26]. It must be highlighted that the conditions for optimal extraction of total phenolics and AA were similar for both cultivars analyzed, despite being of different colors and different morphological characteristics.

It must be stressed that TPC and AA exhibited a positive linear correlation (R^2^ = 0.936 and 0.929 for LR and M, respectively), suggesting that phenolic compounds could be major contributors to AA in rose petals. This trend was also reported for many medicinal plants, vegetables, fruits, and herbs [3,27,28,29].

### 3.3. Experimental Validation

To verify the predictive ability of the model, experimental and predictive responses for 50% ethanol at 90 °C for 30 min extraction for LR were compared. Under these conditions, the experimental and predicted values for TPC were 163.6 ± 2.1 and 163.2 mg GA/g dw, respectively, and those for AA were 463.7 ± 11.4 and 479.9 mg Trolox/g dw, respectively. The mentioned values exhibited differences lower than 4%, confirming that the response surface methodology model proposed is reliable and accurate.

### 3.4. Effect of Ultrasound-Assisted Extraction

The effect of ultrasound on the extraction of total phenolics and AA was evaluated for the selected conditions: 38% ethanol, 75 °C and 30 min (Table 5). This treatment significantly increased the AA in the M and LR hydroethanolic extracts, while no significant differences were found in TPC. Most likely, ultrasound favored the extraction of antioxidant compounds other than phenolics. It is well known that ultrasound produces the collapse of cavitation bubbles with very high energy and speed. This effect could promote the disruption of the stagnant layer of solvent surrounding the extractable material, contributing to the release of antioxidant compounds [30]. In addition, the high shear forces and other physical forces generated by acoustic cavitation enhance mass transfer, increasing the extraction capacity of antioxidant compounds [31]. However, prolonged sonication can degrade some compounds, such as anthocyanin [32]. Therefore, the use of ultrasound-assisted extraction reinforces the previous selection of a shorter extraction time.

### 3.5. Comparison of Drying Methods Using the Selected Conditions

The evaluation of drying methods was carried out with extracts obtained under different conditions from those selected by response surface methodology, since this analysis was prior to the optimization of the extraction. Therefore, both drying methods were also compared in the LR cultivar with the extractions performed under the selected conditions (38% ethanol, 75 °C, 30 min, and ultrasound assistance). The TPC was 160.7 ± 8.6 mg GA/g dw, and the AA was 552.6 ± 23.0 mg Trolox/g dw for extracts from freeze-dried powders. The TPC and AA for extracts from air-dried powders were 164.1 ± 2.0 mg GA/g dw and 548.1 ± 1.52 mg Trolox/g dw, respectively. There were no significant differences in TPC and AA between the freeze-drying and hot air-drying methods under the selected conditions.

### 3.6. HPLC-ESI-QTOF/MS Analysis

Extracts of the LR cultivar obtained under the selected conditions were investigated by HPLC-ESI-QTOF/MS to compare the effect of freeze and hot air-drying methods on their phenolic composition. It should be mentioned that the chromatographic profiles obtained for both hydroalcoholic extracts were coincident, since equal retention times were the same for each peak (Figure 4). Furthermore, there were no significant differences between the areas corresponding to each peak for both treatments. Therefore, and in agreement with the results for total phenols, the drying method did not affect the content of any individual phenolic. Each compound was identified considering the accurate mass measured, the mass fragmentation pattern in negative ionization mode, and data reported in the literature. The tentatively identified compounds with their Rt., molecular formula, *m*/*z*, and MS/MS fragments are shown in Table 6. A total of 17 compounds were observed. Nine of them were flavonol glycosides, five quercetin glycosides, and three kaempferol glycosides, which are known components of rose petals [23,33,34]. The other, a myricetin glycoside, was also detected by Cendrowski et al. in Rosa rugosa petals [35]. Catechin, a flavan-3-ol previously found in rose petals, was observed in peak 2 [35,36]. Four ellagitannins (including two unidentified) were detected in peaks 3, 6, 7, and 8, and ellagic acid was observed in peak 12 according to the literature data [33,34,35,36,37]. Ellagitannins are polymeric structures in the form of glycosides, including galloyl and hexahydroxydiphenoyl (HHDP) units. These valuable compounds exhibit antioxidant, anticancer, and anti-inflammatory features [35]. The malic acid derivative in peak 4 was identified according to Mohsen et al. [34]. In addition, a nonphenolic compound was detected in peak 1, which was previously identified as quinic acid in the species *Rosa damascena* [33,34]. To our knowledge, this is the first report on the phytochemical composition of Argentine rose cultivars.

## 4. Conclusions

Freeze drying and hot air drying were convenient alternatives for preserving the TPC and AA of rose petals. Moreover, neither method affected the phenolic composition differently from the other. However, regarding red color retention, freeze drying was more efficient. The solvent was the factor that most affected the extraction of total phenolics and AA, followed by temperature, while time exhibited a minor effect. The selection of optimal extraction conditions is a key factor in maximizing the recovery of phenolics and AA from rose petals. The use of response surface methodology allowed us to optimize these variables and was also a convenient tool for analyzing the interactions between them. The utilization of ultrasound improved the extraction of antioxidant compounds. The results showed that the Argentinean roses analyzed, particularly the red cultivars, contained an appreciable quantity of phenolic compounds with AA. Thus, they can be considered a promising natural source of valuable compounds, and at the same time, the use of early discarded flowerings would contribute to the reduction of agro-industrial waste.

## Figures and Tables

**Figure 1 antioxidants-12-00681-f001:**
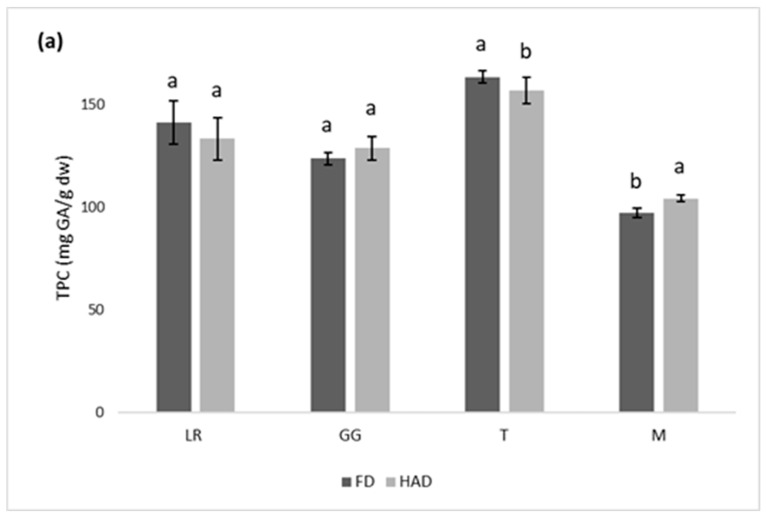
Effect of hot air (HAD) and freeze drying (FD) on (**a**) total phenolic content (TPC), (**b**) antioxidant activity (AA), and (**c**) the colorimetric parameter a* for *Traviata* (T), *Gran Gala* (GG), *Lovely Red* (LR), and *Malu* (M) rose cultivars. Different letters for each cultivar represent significant differences (*p* ≤ 0.05).

**Figure 2 antioxidants-12-00681-f002:**
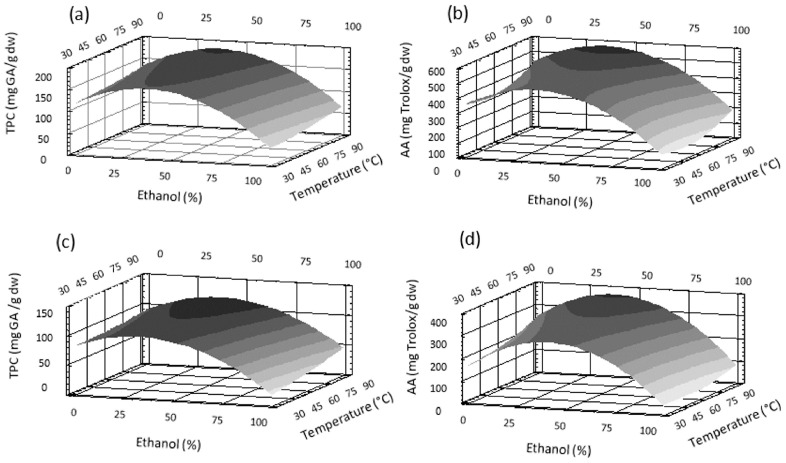
Response surface plots for the effect of ethanol percentage and temperature at 75 min on total phenolic content (TPC) in (**a**) *Lovely Red* and (**b**) *Malu* and on antioxidant activity (AA) in (**c**) *Lovely Red* and (**d**) *Malu*.

**Figure 3 antioxidants-12-00681-f003:**
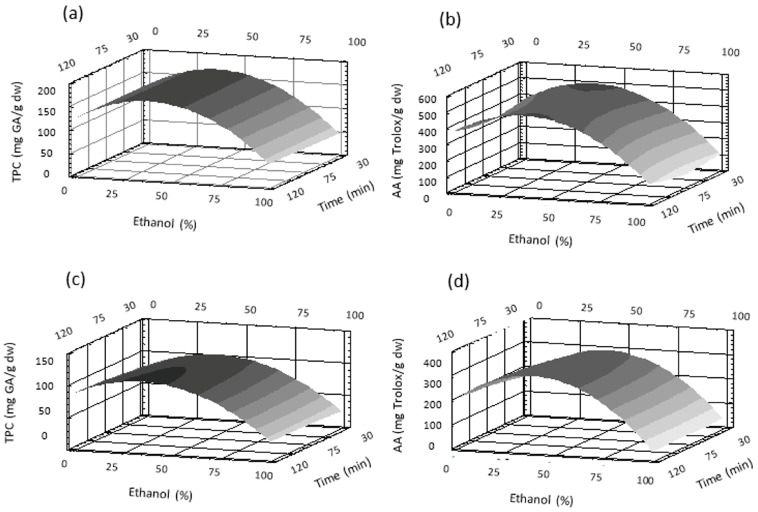
Response surface plots for the effect of ethanol percentage and time at 60 °C on total phenolic content (TPC) in (**a**) *Lovely Red* and (**b**) *Malu* and antioxidant activity (AA) in (**c**) *Lovely Red* and (**d**) *Malu*.

**Figure 4 antioxidants-12-00681-f004:**
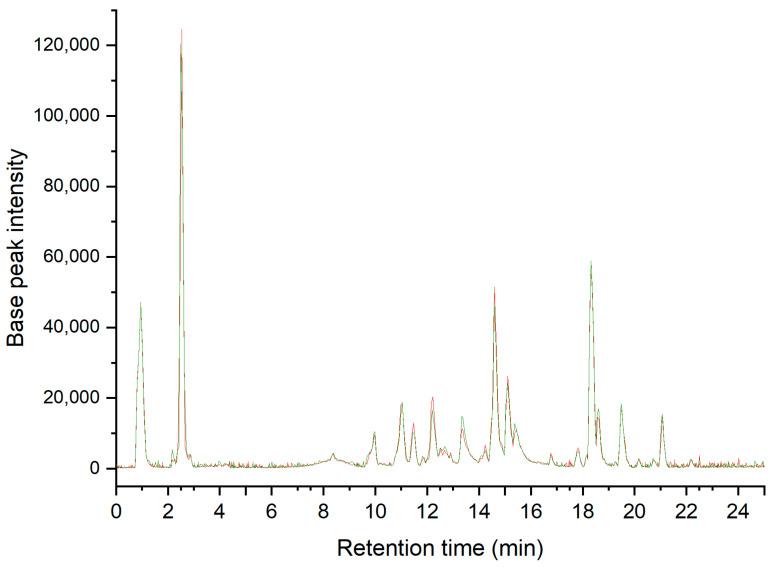
HPLC chromatograms of extracts from freeze-dried (green line) and air-dried (red line) petals of *Lovely Red* cultivar obtained under the selected conditions (38% ethanol, 75 °C, 30 min, with UAE).

**Table 1 antioxidants-12-00681-t001:** Extraction parameters: real and coded values.

Independent Variables		Variables with Their Coded Levels		
		−1	0	1
Ethanol (%)	X_1_	0	50	100
Temperature (°C)	X_2_	30	60	90
Time (min)	X_3_	120	75	30

**Table 2 antioxidants-12-00681-t002:** Box–Behnken design with uncoded levels of each variable and experimentally observed responses of total phenolics and antioxidant activity.

Run	Variable Levels			Total Phenolic Content (mg GA/g dw)		Antioxidant Activity (mg Trolox/g dw)	
	Ethanol (%, X_1_)	Temperature (°C, X_2_)	Time (min, X_3_)	*Lovely Red*	*Malu*	*Lovely Red*	*Malu*
1	100	90	75	75.1 ± 1.2	41.4 ± 0.3	214.1 ± 3.9	83.8 ± 1.2
2	50	30	30	167.2 ± 2.9	83.0 ± 1.4	535.6 ± 5.6	253.3 ± 2.0
3	50	90	120	166.4 ± 3.3	118.4 ± 0.9	581.8 ± 2.8	359.3 ± 4.0
4	100	60	30	46.4 ± 0.8	30.5 ± 0.2	116.5 ± 0.1	26.6 ± 0.0
5	100	30	75	29.1 ± 0.2	15.2 ± 0.1	69.6 ± 2.7	25.5 ± 0.4
6	50	30	120	160.5 ± 2.0	100.6 ± 1.6	522.9 ± 1.6	307.4 ± 1.7
7	0	90	75	148.5 ± 2.3	93.1 ± 1.9	484.4 ± 5.0	267.3 ± 2.1
8	50	90	30	173.9 ± 2.4	114.2 ± 0.7	562.3 ± 5.2	345.3 ± 1.0
9	100	60	120	54.7 ± 0.8	34.9 ± 0.1	138.6 ± 0.7	55.8 ± 0.4
10	0	60	120	130.7 ± 2.3	85.1 ± 0.9	360.1 ± 13.9	204.1 ± 1.2
11	0	30	75	113.1 ± 0.9	87.0 ± 1.3	331.8 ± 8.9	127.9 ± 1.3
12	0	60	30	121.8 ± 1.2	73.4 ± 2.2	370.5 ± 10.3	150.4 ± 10.0
13	50	60	75	163.5 ± 5.9	106.9 ± 1.5	407.0 ± 10.4	307.7 ± 2.0
14	50	60	75	159.5 ± 0.8	92.5 ± 1.9	498.3 ± 2.2	306.8 ± 1.3
15	50	60	75	163.0 ± 1.2	105.4 ± 2.7	499.7 ± 6.0	291.1 ± 7.4
16	100	90	75	72.9 ± 2.5	42.3 ± 03	220.5 ± 3.2	77.4 ± 2.1
17	50	30	30	158.8 ± 4.3	97.0 ± 1.6	524.2 ± 9.4	293.9 ± 5.2
18	50	90	120	173.3 ± 3.5	115.7 ± 0.3	557.2 ± 3.3	338.1 ± 1.8
19	100	60	30	47.0 ± 0.8	29.2 ± 0.2	116.5 ± 0.1	26.5 ± 0.0
20	100	30	75	28.1 ± 1.7	16.5 ± 0.2	65.0 ± 3.9	27.7 ± 1.4
21	50	30	120	162.9 ± 2.4	100.6 ± 1.6	528.0 ± 8.8	264.4 ± 1.8
22	0	90	75	148.2 ± 4.2	95.9 ± 1.0	473.1 ± 10.6	264.2 ± 5.8
23	50	90	30	168.6 ± 3.1	112.8 ± 1.4	550.9 ± 7.9	356.6 ± 3.0
24	100	60	120	51.8 ± 1.3	33.3 ± 0.5	131.9 ± 0.7	53.1 ± 4.1
25	0	60	120	123.4 ± 1.2	84.5 ± 2.9	375.1 ± 6.3	225.8 ± 5.6
26	0	30	75	105.9 ± 2.6	85.8 ± 0.5	332.6 ± 3.5	213.9 ± 0.8
27	0	60	30	126.6 ± 1.7	74.3 ± 0.7	374.1 ± 6	146.2 ± 5.2
28	50	60	75	165.5 ± 1.9	109.4 ± 0.6	419.4 ± 6.3	318.9 ± 3.2
29	50	60	75	167.3 ± 2.4	105.0 ± 0.5	432.3 ± 1.0	299.1 ± 0.8
30	50	60	75	164.9 ± 2.6	105.8 ± 0.5	457.5 ± 11.8	298.5 ± 3.3

n = 3 for experimentally observed values.

**Table 3 antioxidants-12-00681-t003:** ANOVA for the effect of ethanol % (X_1_), temperature (X_2_), and time (X_3_) on total phenolic content (TPC) and antioxidant activity (AA).

Source	DF	Sum of Squares		*F* Ratio		*p* Value	
		*Lovely Red*	*Malu*	*Lovely Red*	*Malu*	*Lovely Red*	*Malu*
TPC							
X_1_	1	23,492.5	11,871.7	325.2	475.58	0.0000 *	0.0000 *
X_2_	1	2535.1	1372.6	35.1	54.97	0.0000 *	0.0000 *
X_3_	1	11.7	215.2	0.2	8.62	0.6925	0.0085 *
X_1_^2^	1	42,902.0	16,376.2	593.9	656.03	0.0000 *	0.0000 *
X_1_X_2_	1	20.0	161.0	0.29	6.45	0.5961	0.0200 **
X_1_X_3_	1	6.8	22.4	0.09	0.90	0.7619	0.3556
X_2_^2^	1	42.5	48.1	0.59	1.93	0.4522	0.1813
X_2_ X_3_	1	0.0018	25.2	0.00	1.01	0.9961	0.3276
X_3_^2^	1	0.09	15.6	0.00	0.63	0.9719	0.4384
Blocks	1	2.3	23.4	0.03	0.94	0.8599	0.3447
Lack of fit	15	1359.6	28.2	28.2	0.66	0.0027	0.7493
Total error	19	1372.4	474.3				
Total (corr.)	29	71,067.5	30,846.9				
R^2^		0.981	0.985				
R^2^ adj		0.972	0.978				
AA							
X_1_	1	260,899	93,535.0	255.02	207.81	0.0000 *	0.0000 *
X_2_	1	33,718.10	20,891.8	32.96	46.42	0.0000 *	0.0000 *
X_3_	1	220.00	2732.15	0.22	6.07	0.6481	0.0235 **
X_1_^2^	1	403,924	254,709	394.83	565.89	0.0000 *	0.0000 *
X_1_X_2_	1	5.80	832.93	0.01	1.85	0.9408	0.1896
X_1_X_3_	1	133.17	751.75	0.13	1.67	0.7222	0.2117
X_2_^2^	1	28,201.3	2392.67	27.57	5.32	0.0000 *	0.0326 **
X_2_X_3_	1	150.43	105.78	0.15	0.24	0.7056	0.6334
X_3_^2^	1	10,439.4	353.09	10.20	0.78	0.0048 *	0.3869
Blocks	1	592.474	281.58	0.58	0.63	0.4560	0.4387
Lack of fit	15	14,076.8	8107.48	0.70	4.86	0.7272	0.0687
Total error	19	19,437.8	8551.95				
Total (corr.)	29	784,957	390,327				
R^2^		0.975	0.978				
R^2^ adj		0.964	0.968				

DF—degrees of freedom. * Highly significant (*p* < 0.01). ** significant (0.01 < *p* < 0.05).

**Table 4 antioxidants-12-00681-t004:** Comparison of predicted values for the response of total phenolic content (TPC) and antioxidant activity (AA) at optimal and other selected conditions of extraction.

Extraction Conditions	Independent Variables			Predictive TPC (mg GA/g dw)		Predictive AA (mg Trolox/g dw)	
	Ethanol (%, X_1_)	Temperature, (°C, X_2_)	Time, (min, X_3_)	*Lovely Red*	*Malu*	*Lovely Red*	*Malu*
Optimal conditions	38	90	117	202.7			
	38	90	112		119.2		
	38	90	120			611.9	
	38	90	114				376.5
Optimal conditions at average time	38	90	116	205.2	119.5	604.8	377.1
Proposed conditions	38	75	30	189.3	108.5	535.6	320.7

**Table 5 antioxidants-12-00681-t005:** Effect of ultrasound-assisted extraction (UAE) with 38% ethanol at 75 °C for 30 min on total phenolic content (TPC) and antioxidant activity (AA).

Response	Cultivar	with UAE	without UAE
TPC (mg GA/g dw)	*Lovely Red*	159.3 ^a^ ± 1.4	160.5 ^a^ ± 6.0
	*Malu*	100.4 ^a^ ± 1.9	100.6 ^a^ ± 2.4
	*Lovely Red*	527.4 ^a^ ± 6.9	481.3 ^b^ ± 10.9
AA (mg Trolox/g dw)	*Malu*	291.9 ^a^ ± 7.7	279.5 ^b^ ± 6.9

n = 6. Different letters in the same row represent significant differences (*p* ≤ 0.05).

**Table 6 antioxidants-12-00681-t006:** Tentative identification of compounds detected in hydroethanolic extracts from rose petals of *Lovely Red* cultivar by HPLC-ESI-QTOF/MS in negative ionization mode.

N°	Rt. (min)	Formula	Compound	[M-H]^−^ (*m*/*z*)	MS/MS
1	2.5	C_7_H_12_O_6_	Quinic acid	191.0558	171, 155, 137, 127
2	10.2	C_15_H_14_O_6_	Catechin	289.0709	179, 145
3	10.8	C_41_H_26_O_26_	Unknown ellagitannin	466.0303 *	301, 451
4	11.2	C_17_H_20_O_14_	Galloyl hexose malic acid	447.0785	331, 169, 133
5	11.8	C_27_H_30_O_18_	Myricetin 3,5-di-O-glucoside	641.1350	541, 479, 410, 355, 317
6	12.2	C_34_H_24_O_22_	Bis-HHDP-hexose	391.0316 *	301
7	12.2	C_41_H_28_O_26_	Galloyl-bis-HHDP-hexose	935.0772	451
8	12.5	C_41_H_26_O_26_	Unknown ellagitannin	466.0291 *	301, 451
9	17.8	C_20_H_18_O_11_	Quercetin-*O*-pentoside	433.0762	410, 300
10	18.1	C_21_H_20_O_12_	Quercetin-*O*-hexoside	463.0899	410, 300
11	18.6	C_14_H_6_O_8_	Ellagic acid	300.9995	283, 257, 245, 229, 210, 201, 185, 173
12	19.3	C_27_H_30_O_16_	Rutin	609.1462	410, 343, 301
13	19.5	C_21_H_20_O_11_	Quercetin-*O*-rhamnoside	447.0930	410, 300, 271, 178, 151
14	19.7	C_28_H_24_O_15_	Quercetin-*O*-galloylrhamnoside	599.1042	447, 410, 313, 285, 226, 169
15	20.2	C_20_H_18_O_10_	Kaempferol-*O*-pentoside	417.0814	284, 255, 227
16	20.7	C_27_H_30_O_15_	Kaempferol-*O*-hexosyl-deoxyhexoside	593.1507	410, 285
17	21.1	C_21_H_20_O_10_	Kaempferol-*O*-deoxyhexoside	431.1003	410, 284, 255, 227

* Corresponds to ion [M-2H]^2−^, HHDP: hexahydroxydiphenoyl.

## Data Availability

The data are contained within the article.

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
