# Peer review of "Effect of the Drying Method and Optimization of Extraction on Antioxidant Activity and Phenolic of Rose Petals"

_antioxidants, 2023, doi:10.3390/antiox12030681_

Round 1
Reviewer 1 Report
The manuscript titled “Effect of the Drying Method and Optimization of Antioxidant Activity and Phenolic Extraction from Rose Petals” could be represent a valid scientific work, but before being considered for publication in this Journal, the authors should make several changes, such as:
- I advise to the authors to revise the title if they consider it appropriate.
- I do not consider the part of the results to be legible and clear. I advice to the authors to clarify the data in a more understandable way.
- You evaluated several extraction methods, but apparently you have only analyzed them for TPC and one antioxidant activity test (I think that is few). Moreover, you analyzed through HPLC-ESI-microTOF/MS the phenolic composition, but the analysis in my opinion is insufficient, should provide more information (such as amount of individual compounds...). In this way, I consider the article not enough to be published in this Journal that deals of Antioxidants.
- Edit references according to Journal style.
Author Response
We are very grateful for your useful comments. We have answered all the questions and corrected the manuscript taking into account your observations. In the cases we did not addressed the comments as recommended, we have explained the reasons.
The responses to the comments or queries are written below (in red) each comment, mentioning the lines with the corresponding corrections in the revised manuscript.
Comments and Suggestions for Authors
The manuscript titled “Effect of the Drying Method and Optimization of Antioxidant Activity and Phenolic Extraction from Rose Petals” could be represent a valid scientific work, but before being considered for publication in this Journal, the authors should make several changes, such as:
Point 1: I advise to the authors to revise the title if they consider it appropriate.
Response 1: We have now changed the title.
Point 2: I do not consider the part of the results to be legible and clear. I advise to the authors to clarify the data in a more understandable way.
Response 2: Some paragraphs in the Results and Discussion section has been improved, as can be seen in lines 210-224, 227-229, 231-234, 358-360, 380, 405 and 413 in the PDF version of the revised Manuscript.
In particular, there were some errors in the SD bars of Fig 1. We have now modified them as well as the letters indicating the significant differences between the drying methods. In accordance with these modifications, we have also changed the comments about these results (see lines 210-211).
We have also included some sentences to clarify the selected conditions (lines 358-360 and 380).
Point 3: You evaluated several extraction methods, but apparently you have only analyzed them for TPC and one antioxidant activity test (I think that is few).
Response 3: The main goal of our study was to obtain extracts from rose petals with the highest possible antioxidant activity to be used as natural antioxidants in foods. Phenolic compounds from flowers are generally recognized as natural antioxidants (Fernandes et al., 2017; Pires et al., 2019). In this sense, the strong effect of anthocyanins has been pointed out (Janarny et al., 2021; Kumari et al., 2021). However, in terms of molar ratio, the proportion of anthocyanins to total phenolics were lower than 1% in the rose cultivars analyzed in our study. Therefore, according to these estimations, the contribution to anthocyanins to the total antioxidant activity in these samples could be considered not relevant. For this reason, we have focused on optimizing the AA and TPC in the extracts. Besides, in 3.1. section, we have included the anthocyanin contents of the four cultivars analyzed and explained the choice of TPC and AA analysis to optimize the extraction conditions (lines 217-224). We have also included in Materials and Method section the determination of anthocyanin content (lines 118-123).
Fernandes, S. Casal, J.A. Pereira, et al. Edible flowers: a review of the nutritional, antioxidant, antimicrobial properties and effects on human health. J. Food Compos. Anal., 60 (2017), pp. 38-50. 10.1016/j.jfca.2017.03.017
T.C.S.P. Pires, L. Barros, C. Santos-Buelga, et al. Edible flowers: emerging components in the diet. Trends Food Sci. Technol, 93 (2019), pp. 244-258, 10.1016/j.tifs.2019.09.020
Janarny, G.; Gunathilake, K.; Ranaweera, K. Nutraceutical potential of dietary phytochemicals in edible flowers-A review. J. Food Biochem. (2021), e13642. 10.1111/jfbc.13642
Kumari, P., Ujala, Bhargava, B. Photochemicals from edible flowers: opening a new arena for healthy lifestyle. J. Funct. Foods, 78, (2021),104375. https://doi.org/10.1016/j.jff.2021.104375.
Regarding the AA, we evaluated its variation according to the different extraction conditions. Therefore, we consider that the utilization of one method should be sufficient for this purpose. The DPPH assay is widely used for the analysis of the antioxidant activity in different matrices, particularly in flowers, with very good results (Fernandes et al., 2017; Liu et al., 2023). Even in some studies it was the only one used (Jagadeesan et al., 2019; Hazar et al., 2023; Mikołajczak et al., 2020; Zemouri‑Alioui et al., 2019). Therefore, we selected this method to assess the antioxidant activity in the different extracts of our study.
Fernandes, S. Casal, J.A. Pereira, et al. Edible flowers: a review of the nutritional, antioxidant, antimicrobial properties and effects on human health. J. Food Compos. Anal., 60 (2017), pp. 38-50. 10.1016/j.jfca.2017.03.017
Xuqiang Liu, Senye Wang, Lili Cui, Huihui Zhou, Yuhang Liu, Lijun Meng, Sitan Chen, Xuefeng Xi, Yan Zhang, Wenyi Kang. Flowers: precious food and medicine resources. Food Science and Human Wellness, 12 (2023), pp 1020-1052. 10.1016/j.fshw.2022.10.022.
Jagadeesan, G., Muniyandi, K., Manoharan, A.L. et al. Optimization of phenolic compounds extracting conditions from Ficus racemosa L. fruit using response surface method. Journal of Food Measurement and Characterization, 13, 312–320 (2019). https://doi.org/10.1007/s11694-018-9946-6
Hazar, D.; Boyar, I.; Dincer, C.; Ertekin, C. Investigation of Color and Bioactive Compounds of Different Colors from Pansy (Viola wittrockiana Gams.) Dried in Hot Air Dryer. Horticulturae 2023, 9, 186. https://doi.org/10.3390/horticulturae9020186
Mikolajczak, D.A. Sobiechowska, M. Tanska. Edible flowers as a new source of natural antioxidants for oxidative protection of cold-pressed oils rich in omega-3 fatty acids. Food Res. Int., 134 (2020), p. 109216
https://doi.org/10.1016/j.foodres.2020.109216
Zemouri-Alioui, S., Bachir bey, M., Kurt, B.Z. et al. Optimization of ultrasound-assisted extraction of total phenolic contents and antioxidant activity using response surface methodology from jujube leaves (Ziziphus jujuba) and evaluation of anticholinesterase inhibitory activity. . Journal of Food Measurement and Characterization, 13, 321–329 (2019). https://doi.org/10.1007/s11694-018-9947-5
We realized that we have not properly explained the aim of our study. Hence, we have now described it in more detail to better convey our idea (lines 77-80 in the revised Manuscript). We have also added a paragraph in the Introduction section related to this purpose (lines 47-50).
Point 4: Moreover, you analyzed through HPLC-ESI-microTOF/MS the phenolic composition, but the analysis in my opinion is insufficient, should provide more information (such as amount of individual compounds).
Response 4: We wanted to compare the effect of freeze and hot air-drying methods on each phenolic composition. Although we did no find significant differences in TPC after both treatments, some components could have shown them. For this purpose, we compared the chromatographic profiles obtained for both hydroalcoholic extracts. If some peaks had been different, we would have quantified the amount of the corresponding compounds to calculate the differences. However, since the chromatographic profiles were coincident, we considered that the quantification was not necessary. We added these chromatograms in the new version of the manuscript (see line 413 and Figure 4).
As can be seen in this same journal, other authors have analyzed the phenolic composition only qualitatively, like Formato et al. (2022), García et al. (2022) or Vargas-Arana et al. (2022).
Formato, M.; Vastolo, A.; Piccolella, S.; Calabrò, S.; Cutrignelli, M.I.; Zidorn, C.; Pacifico, S. Antioxidants in Animal Nutrition: UHPLC-ESI-QqTOF Analysis and Effects on In Vitro Rumen Fermentation of Oak Leaf Extracts. Antioxidants 2022, 11, 2366. https:// doi.org/10.3390/antiox11122366
García, M.C.; Lombardo-Cristina, V.; Marina, M.L. Multifunctional and Collaborative Protection of Proteins, Peptides, Phenolic Compounds, and Other Molecules against Oxidation in Apricot Seeds Extracts. Antioxidants 2022, 11, 2354. https://doi.org/10.3390/antiox11122354
Vargas-Arana, G.; Merino-Zegarra, C.; del-Castillo, Á.M.R.; Quispe, C.; Viveros-Valdez, E.; Simirgiotis, M.J. Antioxidant, Antiproliferative and Anti-Enzymatic Capacities, Nutritional Analysis and UHPLC-PDA-MS Characterization of Ungurahui Palm Fruits (Oenocarpus bataua Mart) from the Peruvian Amazon. Antioxidants 2022, 11, 1598. https://doi.org/10.3390/antiox11081598
Point 5: Edit references according to Journal style.
Response 5: The references were modified according to Journal style.

Reviewer 2 Report
The paper presents the results of research on the influence of the extraction method on the content of total phenolic compounds (TPC) and antioxidant activity (AA) of rose petals from four different cultivars. The obtained results showed that both lyophilization and hot-air drying did not differ significantly when considering TPC and AA. In addition, surface response methodology was used to optimize the extraction conditions. The results showed that the extraction time had the least effect on TFC and AA. HPLC-ESI-microTOF/MS analysis was also used for the preliminary identification of the components of the aqueous ethanol extract - 17 compounds were identified.
The manuscript is interesting, showing rose petal extracts as a rich source of health-promoting compounds. However, authors should clarify at least one point before publication.
As is well known, the antioxidant properties of plant extracts are closely correlated with the content of polyphenols. It follows that the authors chose two closely correlated parameters. Therefore, the numerical values for both parameters changed in a similar way. In my opinion, in addition to TPC, some other parameter should be selected that would not be correlated with TPC, e.g. the content of other secondary metabolites not correlated with TPC.
Author Response
We are very grateful for your useful comments. We have taken into account your observations and we have explained the reasons why we did not addressed the comments as recommended.
The responses are written below the comment (in red), mentioning the lines with the corresponding corrections in the revised manuscript.
Comments and Suggestions for Authors
Point 1: The paper presents the results of research on the influence of the extraction method on the content of total phenolic compounds (TPC) and antioxidant activity (AA) of rose petals from four different cultivars. The obtained results showed that both lyophilization and hot-air drying did not differ significantly when considering TPC and AA. In addition, surface response methodology was used to optimize the extraction conditions. The results showed that the extraction time had the least effect on TFC and AA. HPLC-ESI-microTOF/MS analysis was also used for the preliminary identification of the components of the aqueous ethanol extract - 17 compounds were identified.
The manuscript is interesting, showing rose petal extracts as a rich source of health-promoting compounds. However, authors should clarify at least one point before publication.
As is well known, the antioxidant properties of plant extracts are closely correlated with the content of polyphenols. It follows that the authors chose two closely correlated parameters. Therefore, the numerical values for both parameters changed in a similar way. In my opinion, in addition to TPC, some other parameter should be selected that would not be correlated with TPC, e.g. the content of other secondary metabolites not correlated with TPC.
Response 1: The main goal of our study was to obtain extracts from rose petals with the highest possible antioxidant activity to be used as natural antioxidants in foods. Phenolic compounds from flowers are generally recognized as natural antioxidants (Fernandes et al., 2017; Pires et al., 2019). In this sense, the strong effect of anthocyanins has been pointed out (Janarny et al., 2021; Kumari et al., 2021). However, in terms of molar ratio, the proportion of anthocyanins to total phenolics were lower than 1% in the rose cultivars analyzed in our study. Therefore, according to these estimations, the contribution to anthocyanins to the total antioxidant activity in these samples could be considered not relevant. For this reason, we have focused on optimizing the AA and TPC in the extracts. Besides, in 3.1. section, we have included the anthocyanin contents of the four cultivars analyzed and explained the choice of TPC and AA analysis to optimize the extraction conditions (see lines 217-224 in the PDF version of the revised Manuscript).
Fernandes, S. Casal, J.A. Pereira, et al. Edible flowers: a review of the nutritional, antioxidant, antimicrobial properties and effects on human health. J. Food Compos. Anal., 60 (2017), pp. 38-50. 10.1016/j.jfca.2017.03.017
T.C.S.P. Pires, L. Barros, C. Santos-Buelga, et al. Edible flowers: emerging components in the diet. Trends Food Sci. Technol, 93 (2019), pp. 244-258, 10.1016/j.tifs.2019.09.020
Janarny, G.; Gunathilake, K.; Ranaweera, K. Nutraceutical potential of dietary phytochemicals in edible flowers-A review. J. Food Biochem. (2021), e13642. 10.1111/jfbc.13642
Kumari, P., Ujala, Bhargava, B. Photochemicals from edible flowers: opening a new arena for healthy lifestyle. J. Funct. Foods, 78, (2021),104375. https://doi.org/10.1016/j.jff.2021.104375.
We realized that we have not properly explained the aim of our study. Hence, we have now described it in more detail to better convey our idea (lines 77-80 in the revised Manuscript). We have also added a paragraph in the Introduction section related to this purpose (lines 47-50).

Round 2
Reviewer 1 Report
The authors have corrected and arranged the work according to my suggestions. In my opinion now the paper is ready to be published.